# All-organic polymeric materials with high refractive index and excellent transparency

Jie Zhang[1,6], Tianwen Bai[1,6], Weixi Liu[2,6], Mingzhao Li[3,4], Qiguang Zang[1], Canbin Ye[3,4], Jing Zhi Sun [1], Yaocheng Shi [2], Jun Ling [1], Anjun Qin [3,4] ✉ & Ben Zhong Tang [1,3,4,5] ✉

High refractive index polymers (HRIPs) have drawn attention for their optoelectronic applications and HRIPs with excellent transparency and facile preparation are highly demanded. Herein, sulfur-containing all organic HRIPs with refractive indices up to 1.8433 at 589 nm and excellent optical transparency even in one hundred micrometre scale in the visual and RI region as well as high weight-average molecular weights (up to 44500) are prepared by our developed organobase catalyzed polymerization of bromoalkynes and dithiophenols in yields up to 92%. Notably, the fabricated optical transmission waveguides using the resultant HRIP with the highest refractive index display a reduced propagation loss compared with that generated by the commercial material of SU-8. In addition, the tetraphenylethylene containing polymer not only exhibits a reduced propagation loss, but also is used to examine the uniformity and continuity of optical waveguides with naked eyes because of its aggregation-induced emission feature.

The world moves from the electronic age to the photon era with the rapid development of science and technology[1,2]. In response to this huge shift, optical materials are attracting increasing attentions in recent years[3–6]. To achieve large-scale photonic integration, high refractive index materials play an irreplaceable role because of their compact structure[7,8]. Although the inorganic materials such as semiconductors (e.g. Ge and Si) and chalcogenide glasses possess very high refractive indices ($n \sim 2.0$–4.0), their development has been limited because they are expensive, toxic, inflexible, and difficult to process[9]. Subsequently, organic-inorganic hybrid materials like nanocomposite come into being[10–13], and the incorporation of high-$n$ nanoparticles into polymers has been explored. However, the non-uniformity of the films, high optical loss caused by Rayleigh scattering and complex preparation process could hardly be avoided[14–16]. Therefore, all-organic polymers, which enjoy the advantages of facile preparation, excellent

processibility, cost-effectiveness, light weight, high performance for highly integrated optical components and circuits have been regarded as the promising candidates[7].

In general, the vast majority of organic polymers possess relatively low $n$ values at the range of 1.5–1.6[17,18], which are not suitable for compact photonic integration[16,19,20], and lead to the small photonic bandgap of photonic crystals[21]. Therefore, the design and exploration of high refractive index polymers (HRIPs) with $n \geq 1.7$ has drawn increasing attention for their attractive optoelectronic applications in organic light- emitting diodes (OLEDs), microlens components in both charge coupled devices (CCDs), high performance complementary image sensors (CISs)[22,23], all-polymer optoelectronic devices[24], polymer-based optical waveguides[7], and so on. In recently years, the HRIPs with $n$ values higher than 1.7 have been prepared[16,25–28]. However, most of them containing π-conjugated

[1]MOE Key Laboratory of Macromolecules Synthesis and Functionalization, Department of Polymer Science and Engineering, Zhejiang University, Hangzhou 310027, China. [2]College of Optical Science and Engineering and International Research Center for Advanced Photonics, Zhejiang University, Hangzhou 310058, China. [3]State Key Laboratory of Luminescent Materials and Devices, Guangdong Provincial Key Laboratory of Luminescence from Molecular Aggregates, South China University of Technology, Guangzhou 510640, China. [4]Center for Aggregation-Induced Emission, South China University of Technology, Guangzhou 510640, China. [5]School of Science and Engineering, Shenzhen Institute of Aggregate Science and Technology, The Chinese University of Hong Kong, Shenzhen, Guangdong 518172, China. [6]These authors contributed equally: Jie Zhang, Tianwen Bai, Weixi Liu. ✉e-mail: msqinaj@scut.edu.cn; tangbenz@cuhk.edu.cn

structures show poor solubility, low optical transmittance in the visible region, and harsh preparation procedures, which greatly limit their optical applications[29].

Among the reported HRIPs, sulfur-containing polymers have drawn much attention in recent years[30–35]. Conspicuously, polyimides[36] and poly(arylene sulfide)s[37] with high *n* values were generated. However, the methods for preparing sulfur-containing polymers under mild reaction conditions are limited[16,38–40]. It is worth mentioning that except for red polymers prepared via inverse vulcanization at elevated temperatures[41–43], the *n* value of other polymers are not very distinguished. Thus, the development of HRIPs with excellent optical properties under mild conditions is still challengeable.

Our groups have been working on the development of alkyne-based polymerizations for years[44]. In 2019, we established a powerful transition metal-free polyaddition reaction of bromoalkyne and phenol monomers, and polyethers containing bromo-vinyl groups on their main-chains were yielded[45]. Sulfur and oxygen elements both belong to the group VIA, so thiophenols and phenols both have nucleophilicity. We were curious that if the phenol monomers were changed to the thiophenol ones, could we develop a new powerful polymerization with bromoalkynes and generate functional polymers with interesting property? After experimental and theoretical investigation, the results demonstrated that thiophenols could readily react with bromoalkynes. Unprecedented, a thiol-yne polymerization using organobase of 1,8-diazabicyclo[5.4.0]undec-7-ene (DBU) as catalyst was successfully established and sulfur-containing polymers with high weight-average molecular weights ($M_w$, up to 44500) were produced in excellent yields (up to 99.2%). Notably, this polymerization is a tandem reaction compositing two steps, that is, one thiol group first substituted the bromo group of bromoalkyne and then another added to carbon atom of ethynyl groups in the formed intermediate, which is totally different from the reaction mechanism of bromoalkyne and phenol[45]. More

excitingly, thanks to their containing tremendous sulfur atoms, bromo atoms and phenyl rings and although having no transition metal species, the polymers show high *n* values (up to 1.8433 at 589) with excellent transparency (near 100% transmission even for one hundred micrometre scaled film in the visible-light region). These advantages endow the polymers with low optical loss for the optical waveguide applications. Moreover, the tetraphenylethylene (TPE)-containing polymer showed the unique aggregation-induced emission (AIE) feature, enables it to be used to visualize the uniformity and continuity of optical waveguides by naked eyes.

## Results

### Polymerization

Owing to the mono-functional feature of bromoalkynes, the carbon atom of ethynyl group without connecting with the bromo group could be easily decorated with such moieties as 4-chlorophenyl, 4-methoxylphenyl, 4-bromophenyl and TPE to generate functional monomers of 1a, 1b, 1c and 1d, respectively (Supplementary Fig. 1). To establish the DBU-catalyzed polymerization of bromoalkynes and dithiols, monomers 1a and 2a were used to optimize the reaction conditions. The reaction temperature, monomer concentration, reaction time and solvent were systematically investigated (Supplementary Tables 1–4). Taking both the molecular weights and yields of the products into account, the reaction temperature of 80 °C, the monomer concentration of 0.2 M, the solvent of dimethyl sulfoxide (DMSO) and the reaction time of 4 h were identified as the optimal polymerization conditions.

Using these optimized conditions, various types of bromoalkyne and dithiophenol monomers were polymerized to explore its robustness and universality. The polymerization of 1 with dithiophenols (2a and 2b) performed smoothly, readily producing polymers P1–P5 with high $M_w$ values (up to 44500) in excellent yields (up to 99.2%) (Fig. 1).

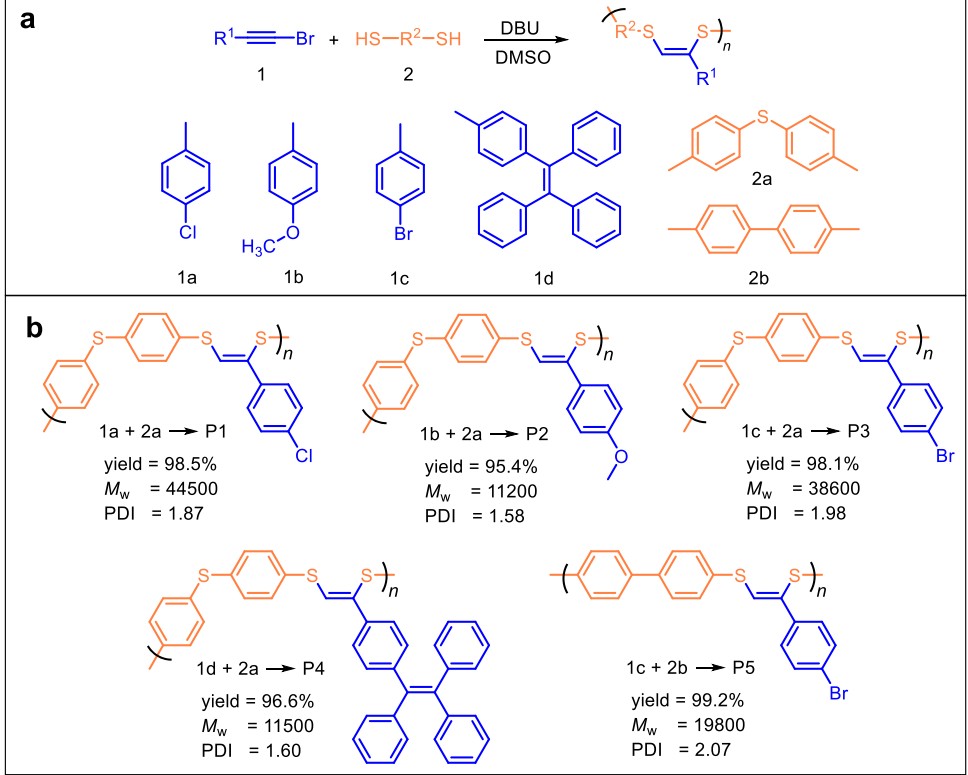

**Fig. 1 | Polymerization and results. a** DBU-catalyzed polymerization of bromoalkynes 1a-1d (blue part) and dithiophenols 2a and 2b (orange part) under nitrogen in DMSO at 80 °C for 4 h ([1]/[2] = 1:1, [1] = 0.2 M, [DBU]/[1] = 1). **b** Chemical structures and polymerization results of P1–P5. $M_w$ and PDI ($M_w/M_n$) of polymers were estimated by GPC in THF on the basis of a polymethyl methacrylate (PMMA) calibration.

The resultant polymers are fully soluble in commonly used organic solvents, such as tetrahydrofuran (THF), dichloromethane, chloroform, toluene, dichlorobenzene, DMSO and dimethylformamide, making them easy to be fabricated into high-quality films. They are also thermally stable. The 5% weight loss temperatures ($T_d$) of all polymers are in the range of 305–347 °C (Supplementary Fig. 2A). Moreover, all polymers have the glass transition temperatures ($T_g$) higher than 92 °C (Supplementary Fig. 2B), making them ideal candidates for optical applications that require shape retention at moderate to elevated temperatures.

Thanks to their excellent solubility, the structures of the polymers were characterized via spectroscopic methods. For the spectral profile of P2 resembles with those of other four polymers, the FT-IR spectra of P2 and its monomers 1b and 2a are shown in Supplementary Fig. 3 as an example. The absorption bands at 2048 and 2544 cm$^{-1}$ are assignable to the stretching vibration of C≡C and S-H in 1b and 2a, respectively. These peaks weakened distinctly in the spectrum of P2, indicative of the reaction of 1b and 2a.

More detailed information of polymers could be obtained from the $^1$H NMR spectra. To facilitate the structural characterization, model compound 4 was prepared under the same reaction conditions (Supplementary Fig. 4). The $^1$H NMR spectra of P2, monomers 1b and 2a, as well as the model compound 4 in CDCl$_3$ are shown in Supplementary Fig. S3 as an example. Characteristic proton c of 4 resonated at δ 7.02 was observed in Supplementary Fig. 5C. Correspondingly, the resonance of proton ć' of P2 was found at δ 7.04 in Supplementary Fig. 5D. Meanwhile, the peak of a' assigned to resonance of methoxyl group was also observed at δ 3.74 in P2. Moreover, the resonant peak associated with thiol group in the spectrum of P2 was hardly observed.

The $^{13}$C NMR spectra were also measured to verify the structures of P2 (Supplementary Fig. 5E–H). The peaks representing carbon atoms d and e in 1b could hardly be found in the spectrum of P2. Meanwhile, the resonance of the carbonyl carbon f in 4 was observed in the spectrum of P2 at δ 113.8. These results again suggested that 1b and 2a had been polymerized, and P2 was successfully yielded. The structural characterization of other polymers resemble that of P2, and their FT-IR, $^1$H and $^{13}$C NMR spectra were shown in Supplementary Figs. 6–8, respectively.

## Reaction mechanism

To clearly figure out the reaction selectivity and fully understand the reaction mechanism, further experimental and theoretical studies were performed. Different from the reaction of bromoalkyne and phenol, in which the hydroxyl group was added to the ethynyl carbon far from the bromo group, the reaction of bromoalkyne and thiophenol was firstly undergone a substitution reaction, that is, the bromo group was substituted by the thiol group, to provide an intermediate 5 (Supplementary Fig. 9) as confirmed by its $^1$H NMR and Mass spectra (Supplementary Figs. 10 and 11). Afterward, the thiol group was added to carbon of ethynyl group. Theoretically, the former could add to both of the carbon atoms of the latter to produce two regioisomers. However, only one isomer that the thiol group added to the carbon atom different from the one that occurred the substitution reaction was yielded as shown by our model reactions (Supplementary Figs. 12 and 13). Thus, we proposed a plausible mechanism for this polymerization. As shown in Fig. 2a, there might be three possible reaction pathways i.e. Z@C3, E@C3 and @C4 when PhS$^-$ attacks different carbons of ethynyl groups. However, experimentally, only one product 6 was obtained, whose purity and structure were unambiguously confirmed by high performance liquid chromatography (Supplementary Fig. 14) and X-ray diffraction analysis (Supplementary Table 5 and Supplementary Data 1), respectively.

To clearly explain the regio- and stereoselectivity of this reaction, DFT calculation was employed. The results showed that $Z$-addition @C3 route is preferred due to lower Gibbs free energy barrier than that of $E$-addition @C3 or @C4 routes ($\Delta\Delta G_1 = 3.54$ and $\Delta\Delta G_2 = 6.51$ kcal/mol, respectively, Fig. 2b), which resulted in a high regioselectivity to produce compound 3. Based on the calculated route, substituent effect was further investigated (Fig. 2c). The DFT calculation revealed that reaction of methoxyl-substituted bromoalkyne and thiophenol had the highest Gibbs free energy barrier and the slowest reaction rate among all cases, and the reactions of Br- and Cl-substituted bromoalkynes and thiophenol showed similar Gibbs free energy barrier and reaction rates. Accordingly, it is understandable that the $M_w$ values of polymers generated from the polymerization of dithiophenol 2a with 1a and 1c are higher than those with 1b and 1d (Fig. 1b).

## Refractive index and optical transparency

As is well known, the introduction of sulfur atoms, aromatic rings and halogen atoms except for fluorine is effective for producing HRIPs[16]. All of these three factors are simultaneously involved in our prepared polymers, thus, they are expected to possess high $n$ values. Indeed, the experimental measurement showed that the $n$ values of all polymers are higher than 1.68 in the whole tested wavelength region (Fig. 3a). Most excitingly, the highest $n$ values ranging from 1.8433 to 1.8023 from 589 nm to 1550 nm was observed for P3. The Abbe's number ($v_D$) of P3 is not high enough, which might be attributed to the absorption of aromatic groups in the near-UV region and it is a general trade-off relationship between $n$ and $v_D$. Notably, P3 possessed a high $v'$ of 125.80 (Table 1), indicative of its low dispersion in NIR region. Moreover, even without any halogen atoms, P4 still retained high $n$ values from 1.7369 to 1.6833 in the same wavelength region. These results indicated that the polymerization of bromoalkynes and dithiophenols is an ideal reaction to generate HRIPs without incorporating any metallic species.

Besides high $n$ values, polymeric materials simultaneously possessing excellent optical transparency are very important for practical applications. Except for the yellowish P4, all other resultant polymers are white powders and could be fabricated into colorless films with high quality by spin-coating technique. So we used P3 with the highest $n$ of 1.8433 to test their transparency by the UV–vis transmission spectroscopy (Fig. 3b). The results showed that near 100% transmission was recorded for nanometer scaled films of P3 in the visible region. It is worth noting that even the film thickness of P3 reached 101 μm (Supplementary Fig. 15), near 100% transmission was still recorded with the wavelength longer than ca. 450 nm. Intuitively, as shown in the inset of Fig. 3b, the background badges and words could be clearly seen through the different thickness films of P3 deposited on glasses. In addition, as C band (1530–1565 nm) is an important and common optical communication band[7], we also tested the transmissivity of their thick films around C band. As shown in Fig. 3c, the results are also very desirable. The excellent transparency and high $n$ values of our polymers are beneficial to be applied in optical field.

## Optical waveguide

Thanks to its high $n$ and excellent transparency, we explored the potential application of P3 in optical transmission waveguides (Fig. 3d, e). For comparison, the waveguides of P4 and conventional SU-8 ($n \sim 1.56$–1.58) were also studied. The propagation losses of the fabricated waveguides were evaluated with different lengths employing the cut-off method (Supplementary Table 6)[46], and the spectra of optical waveguides fabricated with P3, P4 and SU-8 are shown in Fig. 3f. By fitting the length dependent optical powers from the waveguide outputs, the propagation losses of P3 and P4 are 0.645 dB/mm and 0.970 dB/mm at 1550 nm wavelengths,

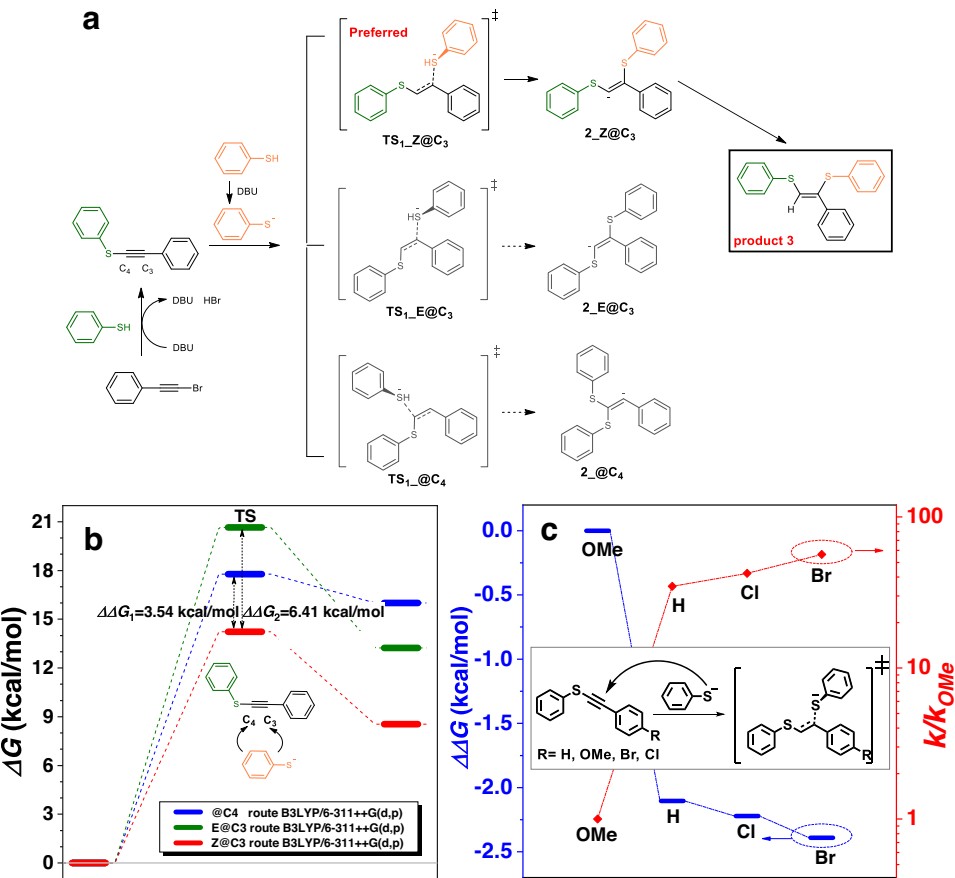

**Fig. 2 | Theoretical investigation of reaction mechanism. a** The possible reaction mechanism of bromoalkyne and thiophenol towards compound **3**. The bromo group was first substituted by the thiol group (olive structure) to provide an intermediate, and then the second thiol group (orange structure) was added to carbon of ethynyl group. There might be three possible reaction pathways i.e. Z@C3, E@C3 and @C4 when PhS⁻ attacked different carbons of ethynyl groups. **b** DFT calculated results of PhS⁻ addition on C3 or C4 of ethynyl group under B3LYP/6-311++G(d,p). *Z*-addition @C3 route (red thick line) is preferred due to lower Gibbs free energy barrier than that of *E*-addition @C3 (olive thick line) or

@C4(blue thick line) routes which resulted in a high regioselectivity to produce compound 3. **c** DFT calculated results of Z@C3 addition with different substituted groups on aromatic ring of alkynes under B3LYP/6-311++G(d,p). The dot line is used as a guide. The blue part refer to the Gibbs free energy and the red part refer to the reaction rate. The reaction of methoxyl-substituted bromoalkyne and thiophenol had the highest Gibbs free energy barrier and the slowest reaction rate among all cases, and the reactions of Br- and Cl-substituted bromoalkynes and thiophenol showed similar Gibbs free energy barrier and reaction rates.

respectively. Notably, compared with the conventional SU-8 waveguide with 1.299 dB/mm loss at 1550 nm, the loss reduction is remarkable because of the higher refractive indices of P3 and P4 than that of SU-8. The insert in Fig. 3f showed the fundamental mode of the polymer waveguide is in the cross-section which is one of the main modes in the waveguide. Moreover, thanks to the containing TPE unit, a typical moiety featuring the aggregation-induced emission (AIE) characteristics, P4 also showed the unique AIE feature (Fig. 4a, b). By taking advantage of this feature, P4 could be filled into the spiral groove of waveguide (Fig. 4c) by spin-coating to visualize its uniformity and continuity by naked eyes upon irradiation by a 445 nm pump light (Fig. 4d). Thereby, our polymers are promising materials for optical device applications.

## Discussion
In summary, a simple and powerful organobase of DBU catalyzed thiol-yne polymerization was successfully established, producing soluble and thermally stable polymers with high $M_w$ values in excellent yields. The resultant polymers contain no metallic species but show high *n* values up to 1.8433 and excellent optical transparency in the visible and NIR region. Benefiting from above advantages, polymeric optical waveguides using a simple fabrication technique were successfully accomplished. The propagation

loss of the fabricated waveguides is remarkably reduced compared with that of SU-8. Thus, our developed thiol-yne polymerization will open up a new avenue for the synthesis of high refractive index polymers, which are promising for applications in optical integrated field.

## Methods
### Materials and instruments
*N*-Bromosuccinimide (>98.0%, T) was recrystallized before use. 4,4'-Thiobisbenzenethiol (>98.0%, GC) and biphenyl-4,4'-dithiol (>98.0%, HPLC) were purchased from TCI, and *p*-toluenethiol (>97.0%, GC) was purchased from Energy. All other chemicals and reagents were purchased from Sigma-Aldrich or Alfa and used as received without further purification.

The weight-average and number-average molecular weights ($M_w$ and $M_n$) and polydispersity indices ($Đ$, $M_w/M_n$) of the polymers were measured by a Waters Advanced Polymer Chromatography (APC) system equipped with a photo-diode array (PDA) detector, using a set of monodisperse polymethyl methacrylate (PMMA) as calibration standards and THF as the eluent in a flow rate of 0.5 mL/min. ¹H and ¹³C NMR spectra were measured on a Bruker AV 400 or Bruker AV 500 spectrometer in deuterated chloroform using tetramethylsilane (TMS; δ = 0) as internal reference at room

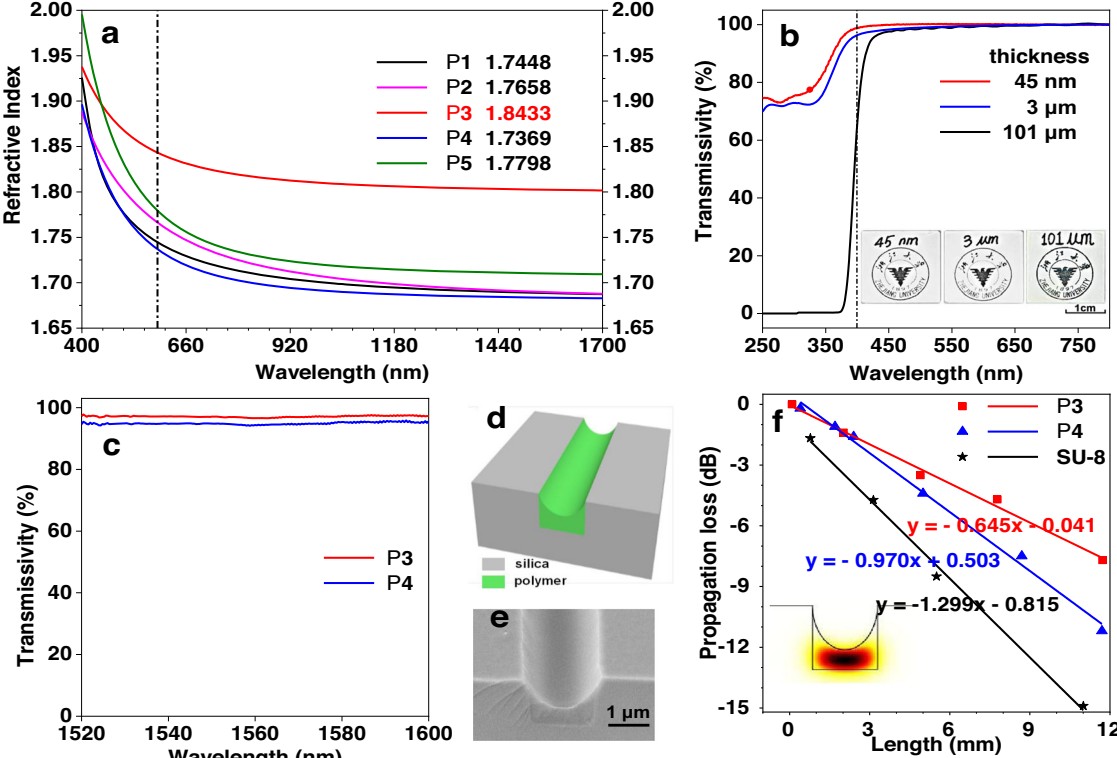

**Fig. 3 | Optical performance and application. a** Light refraction spectra of thin solid films of polymers P1–P5. The *n* values of all polymers are higher than 1.68 in the whole tested wavelength region. And the highest *n* values ranging from 1.8433 to 1.8023 from 589 nm to 1550 nm was observed for P3. **b** UV–vis transmission spectra of **P3** with varying thicknesses (45 nm, 3 μm and 101 μm) on glass showed near 100% transmission with the wavelength longer than 450 nm. The background badges and words could be clearly seen through the different thickness films of P3 deposited on glasses as shown in the inset. **c** Relative IR transmission spectra of P**3** and P**4** in quartz plates showed their excellent transparency. **d** Cross section of the present polymer waveguide. **e** SEM image of the cross section of optical waveguide of P3. **f** The propagation losses of P3, P4 and SU8 at 1550 nm are 0.645 dB/mm, 0.970 dB/mm and 1.299 dB/mm by fitting the length dependent optical powers from the waveguide outputs, respectively. The insert in **f** showed the fundamental mode of the polymer waveguide is in the cross-section which is one of the main modes in the waveguide.

temperature. Fourier transform infrared (FT-IR) spectra were recorded on a Bruker Vector 33 spectrometer as thin films on KBr pellets. Thermal analysis were evaluated on a PerkinElmer TGA 7 and DSC Q20 V24.10 Build 122 under dry nitrogen at a heating rate of 10 °C/min. Refractive indices of the polymers were measured on J. A. Woollam V-VASE variable angle ellipsometry system with a wavelength region from 400 to 1700 nm. The step size was 10 nm, and the incident angle was 65°. The polymer films were prepared by spin coating using 1,2-dichlorobenzene as the solvent on crystalline silicon. The Cauchy dispersion law was applied to analysis the polymer layer from the visible to the IR spectroscopic region. Photoluminescence spectra were recorded on a Horiba Fluoromax-4 spectrofluorometer. The 6 μm thick silica layer on silicon is grown by STS PECVD. The silica groove is etched by STS Multiplex ICP

Etcher. The lithography of positive-tone resist AZ-5214 is made by SUSS-MA6. The propagation loss of polymer waveguide is measured by YOKOGAWA AQ6370. The model number of the 1550 nm lensed fiber is g652d.

### DFT calculation

All geometries of intermediates (Supplementary Fig. 16) were optimized under tight criteria using B3LYP/6-311++G(d,p) method[47,48]. Frequency calculations confirmed that the intermediates on ground state had zero imaginary frequency. Thermal correction to Gibbs free energies was obtained at 298.2 K and 1.013 × 10⁵ Pa. All calculations were performed using Gaussian 16 program, revision B.01[49]. Rate constant was calculated through RRKM method[50].

### Fabrication of waveguides

The solutions of P3 in *o*-dichlorobenzene, SU-8 in cyclopentanone, and P4 in THF/*o*-dichlorobenzene (v/v, 3/7) with concentration 30 mg/mL were filtered by passing through filters (0.22 μm) for three times. Inverted-rib waveguides were fabricated by spin-coating P3, P4, and SU-8 solutions into the pre-etched channels. The fabrication processes is as followings: first, a 6 μm silicon dioxide layer was grown on the silicon substrate; a positive-tone resist AZ-5214 with 1.6 μm thickness was spin-coated on top of the silicon dioxide layer; ultra-violet lithography was used to define the waveguide patterns; inductively coupled plasma reactive-ion-etching was adopted to transfer the pattern from the resist to the silicon dioxide layer with mixed gases of CF₄ and CHF₃ (20 sccm and 36 sccm) to obtain grooves with 3 μm depth and 2 μm width; the

### Table 1 | Refractive indices and chromatic dispersions of P1–P5ᵃ

| Entry | Polymer | $n_D$ | $n_{1550}$ | $u_D$ | $D$ | $u'$ | $D'$ |
|---|---|---|---|---|---|---|---|
| 1 | P1 | 1.7448 | 1.6888 | 12.4083 | 0.0806 | 78.6 | 0.0127 |
| 2 | P2 | 1.7658 | 1.6895 | 11.8913 | 0.0841 | 51.0 | 0.0196 |
| 3 | P3 | 1.8433 | 1.8023 | 19.5208 | 0.0512 | 125.8 | 0.0080 |
| 4 | P4 | 1.7369 | 1.6833 | 10.7661 | 0.0929 | 107.1 | 0.0093 |
| 5 | P5 | 1.7798 | 1.7105 | 8.4630 | 0.1182 | 173.9 | 0.0037 |

ᵃAbbreviation: *n* = refractive index, $u_D$ = Abbé number = $(n_D − 1)/(n_F − n_C)$, where $n_D$, $n_F$, and $n_C$ are the *n* values at wavelengths of 589.3, 486.1, and 656.3 nm, respectively. *D* = chromatic dispersion = $1/u_D$. $u'$ = Abbé number = $(n_{1319} − 1)/(n_{1064} − n_{1550})$, where $n_{1319}$, $n_{1064}$, and $n_{1550}$ are the *n* values at wavelengths of 1319, 1064, and 1550 nm, respectively, $D' = 1/u'$.

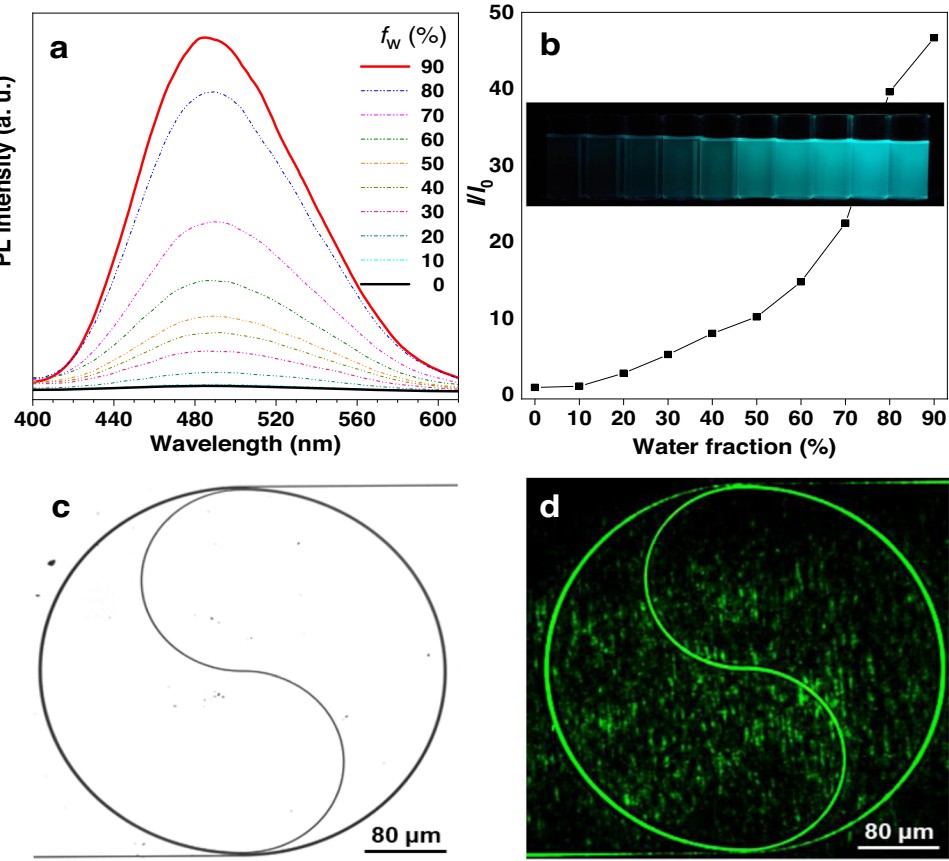

**Fig. 4 | AIE feature of P4 and its application. a** PL spectra of P4 in THF and THF/ water mixtures with different water fractions ($f_w$). Concentration: $10^{-5}$ M; excitation wavelength: 313 nm. **b** The PL intensity enhanced gradually with the water fraction in THF/water mixtures increased, which was visibly showed in the inset. **c** P4 was filled into the spiral groove of waveguide by spin-coating, and the spiral route of waveguide was taken by microscope with a scale bar of 80 μm. **d** By taking advantage of AIE, the uniformity and continuity of the spiral waveguide could be visualized by naked eyes upon irradiation by a 445 nm pump light.

prepared solutions were then drop-casted onto the wafers and spin-coated at a speed of 600 rpm/s. After that, the samples were placed on a nitrogen-filled space for 15 min to ensure adequate backflow of the polymer solutions.

## Data availability

The crystallographic data generated in this study have been deposited in the Cambridge Crystallographic Data Centre database under accession code CCDC 1889353 [https://www.ccdc.cam.ac.uk/ structures/]. The data generated in this study are provided in Supplementary Information. All other data are available from the corresponding authors upon request. Source data are provided with this paper.

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

## Acknowledgements

This work was financially supported by the National Natural Science Foundation of China (21788102 and 21525417), and the Innovation and Technology Commission of Hong Kong (ITC-CNERC14SC01). A.J.Q. and B.Z.T. thank the support from the Natural Science Foundation of Guangdong Province (2019B030301003 and 2016A030312002).

## Author contributions

Monomer design, polymer synthesis, structure characterization, property investigation and manuscript preparation were performed by J.Z. The fabrication and measurement of P3 film with thickness of 101 μm, and the model reaction were done by M.L. Theoretical calculation was performed by T.B. and J.L. Fabrication of waveguides was performed by W.L. Crystallographic data treatments and fluorescent detection were performed by Q.Z. Graphical improvement and data analysis were performed by C.Y. and J.Z.S. Guidance on research and manuscript writing, review, and editing were performed by J.L., Y.S., A.Q. and B.Z.T.

## Competing interests

The authors declare no competing interests.
