## [Peer Review File · Nature Communications]

All-Organic Polymeric Materials with High Refractive Index and Excellent TransparencyReviewers' Comments:

Reviewer #1:

Remarks to the Author:

In this study, based on the bromoalkyne and phenol reaction published in 2019 (Macromolecules 2019, 52, 8, 2949–2955, DOI:10.1021/acs.macromol.9b00306), a transparent high refractive index polymer was claimed to be synthesized through a new chemistry-based click reaction between bromoalkyne and thiophenol. The synthesized polymer has a high refractive index (~ 1.8433 at 589nm), is transparent to both micrometer and visible range, and shows a high yield ($>95\%$), and it has been shown that optical loss is actually reduced by using the developed polymer for an optical waveguide. The evidence of the reaction mechanism presented in the polymer synthesis process presented in this study is not sufficient, and some contradictory parts were also found. It is quite questionable whether the polymer is synthesized as intended. Putting these opinions together, I do not think the manuscript can satisfy the high standard that Nature Communications require. Some comments are attached below.

- There is a lack of theoretical background explanation for the use of thiol instead of phenol in the polymer synthesis. An explanation of the merits and novelty of the proposed chemistry should be further presented.
- It is claimed that the bromo group is substituted first with a thiol group and then the thiol reacts with the carbon on the ethynyl group. However, if bromo is substituted by the thiol group first, the triple bond remains in the structure presented as intermediate 5, but no peak corresponding to the functionality was detected in ^{13}C NMR.
- It was emphasized that only one structure is created among the products that can be created with no clear evidence. The low Gibbs free energy cannot warrant authors' claim. More clear evidences should be presented.
- Under the conditions where R-S minus ions are generated, S-S bonds are easily formed, making it difficult to match the monomer equivalence ratio. The Alkyne side is also not polarized, so it seems at least extremely difficult to act as an electrophile.
- The yield of the unit reactions in Supporting are mostly quite low, but the fact that such high molecular weight polymers are easily made through inefficient reactions with high yield does not conform to the basics of polymer chemistry.
- The explanation for the high refractive index of the developed polymer is not clear. As a reason for the high refractive index, no explanation is given in the manuscript other than that halogen elements have high atomic refraction. However, many polymers using similar elements have been developed so far, and most of them show relatively high values compared to those with values between 1.7 and 1.75, and an additional explanation is needed for this.
- In the case of refractive index, it seems that the refractive index value was obtained using an ellipsometer for the spin-coated polymer film, but no information about the model using the specific sampling and measurement method is shown. It is necessary to present raw data and data fitting process for that part. It is also necessary to show the values in the direction of TE and TM and the refractive index measured directly through the prism coupler measurement, not the fitting data.
- Unlike other polymers synthesized only in the case of P3, the Abbe number is shown to be quite high, which is contrary to the high optical dispersion in the case of polymers containing benzene presented in the intro.
- There are many typos and parts that do not match the figure. For example, there are Figure 5c and

5d in the text, but figure numbering is up to 4. In Figure 1, the monomer notation is incorrect in the equations of reactions P4 and P5.

Reviewer #2:

Remarks to the Author:

This manuscript develops a facile method for preparing high refractive index polymers (HRIPs) and provides five cases of HRIPs. In this work, the authors have carried out a detailed study on the reaction mechanism and the properties of the obtained polymers. Polymer (P3) containing bromine and sulfur shows excellent optical properties, whose refractive index higher than 1.80 even at 1550nm. Thus I recommend acceptance of this manuscript publication if the manuscript will address the following points.

1. There are still mistakes in this article due to negligence. For example: a. "The polymerization of 1 with dithiophenols (2a and 2b) performed smoothly" may be changed to "The polymerization of 1 with dithiophenols (2a and 2b) performed smoothly" b. "Meanwhile, the peak of c' assigned to resonance of methoxyl group was also observed at δ 3.74 in P2" should be changed to "Meanwhile, the peak of a' assigned to resonance of methoxy group was also observed at δ 3.74 in P2"; c. "By taking advantage of this feature, P4 could be filled into the spiral groove of waveguide (Figure 5c)" should be changed to "By taking advantage of this feature, P4 could be filled into the spiral groove of waveguide (Figure 4c)".

2. The thickness of a film sample used for the measurement of the transparency is only 3 μm , which does not reflect the transparency of the film very well. It is suggested that the authors change the thickness of the film to at least 30 μm .

3. The authors have omitted the important references that are regarding to the refractive index polymers based on the thio-ene reaction. Therefore, the authors should quote the following references: a. *Macromolecules* 2018, 51, 7567-7573; b. *Macromolecules* 2020, 53, 125-131; c. *Mater. Chem. Front.* 2021, 5, 5826-5832.

Reviewer #3:

Remarks to the Author:

The manuscript submitted by Zhang et al. is of good scientific quality. In the abstract and in the body of the text authors claim that they developed a thiol-yne polymerization of HRIP polymers. The authors should consider 'High refractive Index Photopolymers by Thiol-Yne "Click" Polymerization' by Mavila et al. (DOI: /10.1021/acsami.1c00831) The use of DBU is well-known for similar reactions. I recommend to express carefully the novelty of the reaction.

The Abbe numbers in the visible range are low (Table 1) and should be referred to as such. As the authors probably look more on the IR range their classification is correct for this wavelength range. in Fig. 1 caption '[1] = 0.2' probably should read '[1] = 2.0' to matched the description in the body of the text.

In caption of Fig. 2c the term 'none' should be replaced by 'H'.

Fig. 4 is not mentioned in the manuscript.

On page 12 the preparation of the spin-coated film contains some duplication (... with 1.6 μm thickness ...)

In case DSC and in particular TGA data are available they should be included.

The chemical novelty of the manuscript is acceptable. Minor spell check is recommended. The manuscript may be published after revision.

Dear Reviewers,

Thank you very much for spending your precious time to review our manuscript and offering us with constructive suggestions to improve the quality of our manuscript. We have very carefully revised our manuscript accordingly.

Below are our specific point-by-point responses to your comments and suggestions.

Response to the comments and suggestions of Reviewer 1

The reviewer commented that “In this study, based on the bromoalkyne and phenol reaction published in 2019 (Macromolecules 2019, 52, 8, 2949–2955, DOI:10.1021/acs.macromol.9b00306), a transparent high refractive index polymer was claimed to be synthesized through a new chemistry-based click reaction between bromoalkyne and thiophenol. The synthesized polymer has a high refractive index (~1.8433 at 589 nm), is transparent to both micrometer and visible range, and shows a high yield (>95%), and it has been shown that optical loss is actually reduced by using the developed polymer for an optical waveguide. The evidence of the reaction mechanism presented in the polymer synthesis process presented in this study is not sufficient, and some contradictory parts were also found. It is quite questionable whether the polymer is synthesized as intended. Putting these opinions together, I do not think the manuscript can satisfy the high standard that *Nature Communications* require. Some comments are attached below.”

We sincerely thank the reviewer for carefully reviewing our manuscript and for raising these good questions and suggestions. We believed that we have fully clarified the reaction and its mechanism by responding the comments and suggestions of the reviewer, and there is no question that the polymers have been synthesized and structurally characterized.

1. *The reviewer commented that “There is a lack of theoretical background explanation for the use of thiol instead of phenol in the polymer synthesis. An explanation of the merits and novelty of the proposed chemistry should be further presented.”*

Response: We thank the reviewer very much for this nice comment and suggestion. Sulfur and oxygen elements both belong to the group VIA, so thiophenols and phenols both have nucleophilicity. Inspired by the success of the polymerization of phenols and bromoalkynes, the reaction of thiophenols with bromoalkynes has been tried. After experimental and theoretical investigation, thiophenols were found capable to react with bromoalkynes indeed, and a clear reaction mechanism with substitution reaction first and followed by an addition

reaction was unambiguously revealed. Thus, the polymerization of dithiophenols with dibromoalkynes has been successfully established. Notably, the reaction mechanism in this work is totally different from that of phenols and bromoalkynes, in which only the addition reaction was occurred and no substitution reaction happened. We have addressed this issue in our revised manuscript.

- The reviewer commented that “It is claimed that the bromo group is substituted first with a thiol group and then the thiol reacts with the carbon on the ethynyl group. However, if bromo is substituted by the thiol group first, the triple bond remains in the structure presented as intermediate **5**, but no peak corresponding to the functionality was detected in ^{13}C NMR.”

Response: We thank the reviewer for raising this question. When carefully surveying the ^{13}C NMR spectrum, we can find that the resonance of the carbons of ethynyl groups could be observed in fact. As shown in Figure C1 (Figure S7 in our SI), the peaks at 77.82 ppm (a) and 95.99 ppm (b) in ^{13}C NMR are readily assignable to the resonance of the ethynyl carbons of intermediate **5**. In addition, the mass spectrum of intermediate **5** was tested by TOF Mass spectrometry. As shown in Figure C2 (Figure S8 in SI), the experimental data of 301.9767 matched well with the theoretical value of 301.9765. These results unambiguously demonstrate the formation of **5** and reasonability of the reaction mechanism. And the corresponding spectra were added in our revised Supplementary Information.

Figure C1. ^{13}C NMR spectrum of intermediate **5** in CDCl_3 .

Figure C2. Mass spectrum of intermediate **5**.

- The reviewer commented that “It was emphasized that only one structure is created among the products that can be created with no clear evidence. The low

Gibbs free energy cannot warrant authors' claim. More clear evidences should be presented."

Response: We thank the reviewer very much for thoughtful comment and suggestion. To prove the sole product of model reaction (Scheme S3), the reaction mixture was just extracted with DCM and alkaline water. The DCM layer was concentrated, and then the crude product was dissolved in hexane. The hexane solution of crude product was detected by high performance liquid chromatography (HPLC) using Waters Alliance e2695 separation module. The HPLC spectra measured with 320 nm UV light detector and 254 nm UV light detector both showed one sole peak (Figure C3) (Figure S10 in SI). These results suggest that only one product was generated and the reaction is quite efficient. These spectra were added in our revised Supplementary Information.

Figure C3. HPLC spectra of the model crude product with detector of 320 nm UV light (A) and 254 nm UV light (B).

4. *The reviewer commented that "Under the conditions where R-S minus ions are generated, S-S bonds are easily formed, making it difficult to match the monomer equivalence ratio. The Alkyne side is also not polarized, so it seems at least extremely difficult to act as an electrophile."*

Response: We thanks the reviewer for this nice comment. In our work, all the reactions were performed under nitrogen, thus, the generated R-S minus ions could not be oxidized by oxygen to form S-S bonds. Meanwhile, the polarization of ethynyl group can be estimated by atom charges. According to DFT calculations, the Mulliken charges of the carbons on ethynyl groups (Chart C1) is 0.149 (C4) and 1.274 (C3), which indicates an obvious polarization of this bond and agrees well with the regio-selectivity in reaction of R-S minus ions.

Chart C1. DFT calculated Mulliken charges of ethynyl group under

B3LYP/6-311++G(d,p).

5. *The reviewer commented that “The yield of the unit reactions in Supporting are mostly quite low, but the fact that such high molecular weight polymers are easily made through inefficient reactions with high yield does not conform to the basics of polymer chemistry.”*

Response: We thank the reviewer for this thoughtful comment. We are sorry that the description of yields in Supplementary Information is unclear and un-specific. The yield of 83.5% for the model compound **6** in the Supplementary Information. is an isolated value after post-processing, including the extraction and purification by column chromatography. This result indicates the reaction is quite efficient and suitable for being developed into a polymerization. Meanwhile, the yield of intermediate **5** is low because the addition reaction will immediately occurred after the first step of substitution reaction.

6. *The reviewer commented that “The explanation for the high refractive index of the developed polymer is not clear. As a reason for the high refractive index, no explanation is given in the manuscript other than that halogen elements have high atomic refraction. However, many polymers using similar elements have been developed so far, and most of them show relatively high values compared to those with values between 1.7 and 1.75, and an additional explanation is needed for this.”*

Response: We thank the reviewer for this good comments. As discussed in the Perspective article published in *Macromolecules* (2015, 48, 1915–1929) by Prof. Mitsuru Ueda: “Obviously, the introduction of aromatic rings, halogen atoms except for fluorine, and sulfur atoms possessing a high atomic refraction is effective for producing high-*n* polymers.” Our prepared polymers contain all of these elements, *i.e.* polarizable aromatic rings, bromo atoms and sulfur atoms, which will contribute to their high refractive index values. Moreover, the linear structure of the polymers might also have contribution. The discussions of “As is well known, the introduction of sulfur atoms, aromatic rings and halogen atoms except for fluorine is effective for producing high-*n* polymers.¹⁶ All of these three factors are simultaneously involved in our prepared polymers, thus, they are expected to possess high *n* values.” were added in our revised manuscript.

7. *The reviewer commented that “In the case of refractive index, it seems that the refractive index value was obtained using an ellipsometer for the spin-coated polymer film, but no information about the model using the specific sampling and measurement method is shown. It is necessary to present raw data and data fitting*

process for that part. It is also necessary to show the values in the direction of TE and TM and the refractive index measured directly through the prism coupler measurement, not the fitting data.”

Response: We thank the reviewer very much for this valuable suggestion. The refractive indices of the polymers were measured on J. A. Woollam V-VASE variable angle ellipsometry system with a wavelength region from 400 to 1700 nm. The step size was 10 nm, and the incident angle was 65°. The Cauchy dispersion law was applied to analysis the polymer layer from the visible to the IR spectroscopic region. The raw data and data fitting process of P1-P5 have been provided in the attachments. The measurement details were given in our revised Supplementary Information.

8. *The reviewer commented that “Unlike other polymers synthesized only in the case of P3, the Abbe number is shown to be quite high, which is contrary to the high optical dispersion in the case of polymers containing benzene presented in the intro.”*

Response: We thank the reviewer very much for this good comment. In RI region, the revised Abbe numbers of polymer are high. However, the ν_D values of polymers are not high. We have revised our description to “The Abbe’s number (ν_D) of P3 is not high, which might be attributed to the absorption of aromatic groups in the near-UV region and it is a general trade-off relationship between n and ν_D . Notably, P3 possessed a high ν' of 125.80 (Table 1), indicative of its low dispersion in NIR region.” in our manuscript. The Abbe numbers of polymers with n values exceeding 1.7 are usually below 20, attributed to the absorption of aromatic groups of these polymers in the near-UV region. Indeed, it is a general trend between n and ν_D that high n and low ν_D (high optical dispersions) are almost coinstantaneous for most optical polymers.

9. *The reviewer pointed out that “There are many typos and parts that do not match the figure. For example, there are Figure 5c and 5d in the text, but figure numbering is up to 4. In Figure 1, the monomer notation is incorrect in the equations of reactions P4 and P5.”*

Response: We thank the reviewer very much for pointing out the typos and mistakes. The “Figure 5c and 5d” have been corrected to “Figure 4c and 4d” in our revised version. And the monomer notations have been corrected with highlighting as followed (Figure C4) (Figure 1 in our revised manuscript).

Figure C4. (a) DBU catalyzed polymerization of bromoalkynes **1a-1d** and dithiophenols **2a** and **2b** under nitrogen in DMSO at 80 °C for 4 h ($[1]/[2] = 1:1$, $[1] = 0.2$ M, $[\text{DBU}]/[1] = 1$). (b) Chemical structures and polymerization results of **P1-P5**. M_w and PDI (M_w/M_n) of polymers were estimated by GPC in THF on the basis of a polymethyl methacrylate calibration.

Response to the comments and suggestions of Reviewer 2

The reviewer commented that “This manuscript develops a facile method for preparing high refractive index polymers (HRIPs) and provides five cases of HRIPs. In this work, the authors have carried out a **detailed study on the reaction mechanism** and the properties of the obtained polymers. Polymer (P3) containing bromine and sulfur shows excellent optical properties, whose refractive index higher than 1.80 even at 1550nm. **Thus I recommend acceptance of this manuscript publication** if the manuscript will address the following points.”

We sincerely thank the reviewer for his/her high appreciation of our work and for his/her constructive suggestions.

1. The reviewer pointed out that "There are still mistakes in this article due to negligence. For example: a. “The polymerization the 1 with dithiophenols (2a and 2b) performed smoothly” may be changed to “The polymerization of 1 with dithiophenols (2a and 2b) performed smoothly” b. “Meanwhile, the peak of c' assigned to resonance of methoxyl group was also observed at δ 3.74 in P2” should be changed to “Meanwhile, the peak of a' assigned to resonance of methoxy group was also observed at δ 3.74 in P2”; c. “By taking advantage of this feature, P4 could be filled into the spiral groove of waveguide (Figure 5c)” should be changed to “By taking advantage of this feature, P4 could be filled into the spiral groove of waveguide (Figure 4c)”.

Response: We thank the reviewer very much for pointing out these mistakes and providing us with the corrections. We have corrected them and carefully checked whole text in our revised manuscript.

- The reviewer commented that “The thickness of a film sample used for the measurement of the transparency is only 3 μm , which does not reflect the transparency of the film very well. It is suggested that the authors change the thickness of the film to at least 30 μm .”

Response: We thank the reviewer very much for this valuable suggestion. We agree with the reviewer that a thicker film than 3 μm will not reflect the transparency very well. Thus, we fabricated a film of P3 with thickness of 101 μm . The results indicated that the transparency of this film is also very good as shown in the Figure C5 (Figure 3b in our revised manuscript). The background badges and words under the film still could be clearly seen. Moreover, the transmission spectrum measurement also show that these films possess good transparency.

Figure C5b. UV-vis transmission spectra for films of P3 with varying thicknesses on glass. Inset: digital images of different thickness of P3 films on glasses.

- The reviewer commented that “The authors have omitted the important references that are regarding to the refractive index polymers based on the thio-ene reaction. Therefore, the authors should quote the following references: a. *Macromolecules* 2018, 51, 7567-7573; b. *Macromolecules* 2020, 53, 125-131; c. *Mater. Chem. Front.* 2021, 5, 5826-5832.”

Response: We thank the reviewer for this valuable suggestion and providing these reference, which have been cited in our revised manuscript.

Response to the comments and suggestions of Reviewer 3

The reviewer commented that “The manuscript submitted by Zhang et al. is of good scientific quality. The chemical novelty of the manuscript is acceptable. Minor spell check is recommended. The manuscript may be published after revision.”

We sincerely thank the reviewer for his/her high appreciation of our work and for his/her constructive suggestions.

1. The reviewer commented that “In the abstract and in the body of the text authors claim that they developed a thiol-yne polymerization of HRIP polymers. The authors should consider 'High refractive Index Photopolymers by Thiol-Yne "Click" Polymerization' by Mavila et al. (DOI: /10.1021/acsami.1c00831).”

Response: We thank the reviewer for this nice advice. This important reference has been cited in our revised manuscript. In this reference, the radical-mediated thiol-yne click polymerization concluded two steps of addition reactions, and is a good method for the preparation of sulfur-containing HRIPs.

2. The reviewer commented that “The use of DBU is well-known for similar reactions. I recommend to express carefully the novelty of the reaction.”

Response: We thank the reviewer very much for this constructive suggestion. DBU is an organic base, and it plays important roles in two aspects of this reaction (Figure C6). In the first aspect, the reaction of bromoalkyne and thiophenol was undergone a **substitution reaction** where the HBr was neutralized by DBU, to provide an intermediate. In the second aspect, the thiol group became R-S minus ions in the presence of DBU, which sequentially added to the carbon atom of ethynyl group of the intermediate. In substance, DBU acts as a de-acid reagent in both steps.

Although the use of DBU is common, this model reaction is new in organic chemistry. What’s more, the polymerization based on this reaction is firstly established, producing a series of polymers with novel structures and excellent properties.

Scheme C6. The model reaction in the presence of DBU.

3. The reviewer commented that “The Abbe numbers in the visible range are low (Table 1) and should be referred to as such. As the authors probably look more on the IR range their classification is correct for this wavelength range.”

Response: We thank the reviewer very much for this valuable advice. We agree with the reviewer’s comment. The v_D values of polymers with n values exceeding 1.7 are usually below 20, attributed to the absorption of aromatic groups of these polymers in the near-UV region. Indeed, it is a general trend between n and v_D that high n and low v_D (high optical dispersions) are almost coinstantaneous for most

optical polymers. On the bright side, the v' values of polymers on the RI range are high. We sincerely took the reviewer's advice and revised our manuscript.

4. The reviewer pointed out that “(1) in Fig. 1 caption '[1] = 0.2' probably should read '[1] = 2.0' to matched the description in the body of the text. (2) In caption of Fig. 2c the term 'none' should be replaced by 'H'. (3) Fig. 4 is not mentioned in the manuscript. (4) On page 12 the preparation of the spin-coated film contains some duplication (... with 1.6 μm thickness ...).”

Response: We thank the reviewer for the nice suggestions. (1) The concentration of monomer **1** should be 0.2 M, which has been corrected in our revised manuscript; (2) The term 'none' has been replaced by 'H' in our revised manuscript; (3) The Fig. 4 has been mentioned in the part of Optical Waveguide before CONCLUSION in our revised manuscript; (4) The repetitive sentence has been removed in our revised manuscript.

5. The reviewer commented that “In case DSC and in particular TGA data are available they should be included.”

Response: We thank the reviewer for this good suggestion. The DSC and TGA curves of polymers are given in Figure C7 (Figure S1 in our revised Supplementary Information). We also discussed the thermal stability of polymers in our revised manuscript as: They are also thermally stable. The 5% weight loss temperatures (T_d) of all polymers are in the range of 305–347 $^{\circ}\text{C}$ (Figure S1A). Moreover, all polymers have the glass transition temperatures (T_g) higher than 92 $^{\circ}\text{C}$ (Figure S1B), making them ideal candidates for optical applications that require shape retention at moderate to elevated temperatures.

Figure C7. (A) TGA and (B) DSC curves of P1-P5. T_d represents the temperature of 5% weight loss. T_g represents glass-transition temperature.

Thank you again for your review.

Reviewers' Comments:

Reviewer #1:

Remarks to the Author:

Authors addressed some issues raised by reviewers and now the revised manuscript became clearer. Now I acknowledge that a new synthetic scheme was suggested and a decently high refractive index polymer was realized therefrom. However, I still do not think that the optical performance is quite impressive. Many of sulfur-containing polymers with the refractive index even higher than 1.9 were reported many times (for example, ACS Macro Lett. 2017, 6, 500–504; ACS Appl. Mater. Interfaces 2021, 13, 61629–61637; Adv. Optical Mater. 2023, 2202432). Some of them had also shown full transparency in visible region (400 - 800 nm) as well. The trade-off relationship between the refractive index and Abbe number still remained unsolved. Authors must clearly address this issue related with the excellency of the optical performance of the P3 polymer compared to other previously reported works. Otherwise, I believe the manuscript should be suitable for other journals specialized in the organic synthesis of polymer materials.

Reviewer #2:

Remarks to the Author:

The authors have revised their manuscript according to the suggestions from the reviewer, I thereby recommend acceptance of this manuscript for publication in NC.

Reviewer #3:

Remarks to the Author:

The authors responded to the reviewers comments and added important informations. In addition the reflection of literature has improved. Still the novelty over the state-of-the-art is not tremendous. But the revised manuscript of good quality may be published as is

Thank you very much for spending your precious time to review our manuscript again. Below are our responses to your comments.

Response to the comments and suggestions of Reviewer 1

The reviewer commented that "Authors addressed some issues raised by reviewers and now the revised manuscript became clearer. Now I acknowledge that a new synthetic scheme was suggested and a decently high refractive index polymer was realized therefrom. However, I still do not think that the optical performance is quite impressive. Many of sulfur-containing polymers with the refractive index even higher than 1.9 were reported many times (for example, *ACS Macro Lett.* 2017, 6, 500-504; *ACS Appl. Mater. Interfaces* 2021, 13, 61629-61637; *Adv. Optical Mater.* 2023, 2202432). Some of them had also shown full transparency in visible region (400-800 nm) as well. The trade-off relationship between the refractive index and Abbe number still remained unsolved. Authors must clearly address this issue related with the excellency of the optical performance of the P3 polymer compared to other previously reported works. Otherwise, I believe the manuscript should be suitable for other journals specialized in the organic synthesis of polymer materials."

Response: We sincerely thank the reviewer for re-reviewing our manuscript and for providing these references. We have perused these papers and cited them in our revised manuscript. As the reviewer commented: the polymerization of sulfur and vinyl groups could generate polymers with higher refractive indices than 1.9. However, two works published in *ACS Macro Lett.* (2017, 6, 500-504) and *Adv. Opt. Mater.* (2023, 2202432) demonstrated that the polymers show transparency in IR instead of visible region. The paper published in *ACS Appl. Mater. Interfaces* (2021, 13, 61629-61637) did show that the polymer is transparent in visible region. However, the preparation of this polymer was described as "Heating elemental sulfur to 190 °C under reduced pressure generates vaporized sulfur radicals by thermal homolytic ring-opening of the elemental sulfur.²³ The presence of a hot filament at 350 °C induces further S-S bond dissociation of the sulfur radicals, leading to the formation of linear short polysulfide radicals in vapor phase.^{24,25} The short polysulfide radicals are mixed homogeneously with the vaporized DVB injected into the sCVD chamber through a separate channel, and free radical polymerization is initiated between the vinyl groups in DVB and sulfur radicals in vapor phase, leading to the deposition of pSDVB-sCVD thin film without any residual elemental sulfur on the target substrate surface." It is concluded that this polymerization and the inverse vulcanization both need higher temperatures than 160 °C and the highest one could reach 350 °C. While, our polymers could be prepared under a milder condition with reaction temperature of

80 °C in organic solvent, which is much lower than aforementioned ones, greatly facilitating their applications.

We fully agree with the reviewer that “the trade-off relationship between the refractive index and Abbe number still remained unsolved.” It has been reported that high- n materials generally exhibit small Abbe’s numbers (Yang, C.-J.; Jenekhe, S. A. Chem. Mater. 1995, 7, 1276–1285). For example, the Abbe’s number of the polymer with sulfur content of 87% (pSDVB-sCVD) was calculated to be 13.83 (ACS Appl. Mater. Interfaces, 2021, 13, 61629-61637), which is lower than that of P3 (19.52). It is our next goal to prepare polymers with high n values and Abbe numbers by fine-tuning the monomeric structures in the future.

For the application of optical materials, the excellency of the optical performance is a key issue but not the only decisive one. The facile preparation, the thermal stability and chemo-stability also need to be taken into account. Compared with the preparation of sulfur-containing polymers, our polymers could be synthesized under a lower reaction temperature. Moreover, our polymers contain no S-S bond, which might be dissociated under reduced condition, enable them to show high chemostability. In conclusion, P3 is processible, and could be prepared by our developed polymerization under a milder reaction conditions at a lower temperature of 80 °C and shows good comprehensive performance including colorless, high RI, excellent transparency, high T_g , and good thermo- and chemo-stability. These advantages of P3 might endow it with wide applications.

Response to the comment of Reviewer 2

The reviewer commented that “The authors have revised their manuscript according to the suggestions from the reviewer, I thereby recommend acceptance of this manuscript for publication in NC.”

Response: We sincerely thank the reviewer for his/her high appreciation and approval of our work.

Response to the comment of Reviewer 3

The reviewer commented that “The authors responded to the reviewers comments and added important informations. In addition the reflection of literature has improved. Still the novelty over the state-of-the-art is not tremendous. But the revised manuscript of good quality may be published as is.”

Response: We sincerely thank the reviewer for his/her appreciation of our work and responses. In this work, we established a novel and simple polymerization with a clear reaction mechanism, producing a series of polymeric materials with clear structures

and good comprehensive performance. Meanwhile, the application of P3 in optical waveguide has showed unlimited potential.

Thank you again for your review.